# Neoadjuvant Therapy Is Associated with Improved Chemotherapy Delivery and Overall Survival Compared to Upfront Resection in Pancreatic Cancer without Increasing Perioperative Complications

**DOI:** 10.3390/cancers14030609

**Published:** 2022-01-26

**Authors:** Christopher Ryan Deig, Thomas Lee Sutton, Blake Beneville, Kristin Trone, Amanda Stratton, Ali N. Gunesch, Amy Ivy Liu, Alaaeddin Alrohaibani, Maedeh Mohebnasab, Solange Bassale, Alison Grossblatt-Wait, Dove Keith, Fouad Attia, Erin W. Gilbert, Charles D. Lopez, Adel Kardosh, Emerson Y. Chen, Kenneth G. Bensch, Nima Nabavizadeh, Charles R. Thomas, Skye C. Mayo, Brett C. Sheppard, Aaron Grossberg

**Affiliations:** 1Department of Radiation Medicine, Oregon Health & Science University, Portland, OR 97239, USA; deig@ohsu.edu (C.R.D.); nabaviza@ohsu.edu (N.N.); 2Brenden-Colson Center for Pancreatic Care, Oregon Health & Science University, Portland, OR 97239, USA; suttoth@ohsu.edu (T.L.S.); grossbla@ohsu.edu (A.G.-W.); keithd@ohsu.edu (D.K.); gilberte@ohsu.edu (E.W.G.); lopezc@ohsu.edu (C.D.L.); sheppard@ohsu.edu (B.C.S.); 3Department of Surgery, Oregon Health & Science University, Portland, OR 97239, USA; trone@ohsu.edu (K.T.); attia@ohsu.edu (F.A.); 4School of Medicine, Oregon Health & Science University, Portland, OR 97239, USA; benevill@ohsu.edu (B.B.); strattoa@ohsu.edu (A.S.); gunesch@ohsu.edu (A.N.G.); 5College of Osteopathic Medicine, Nova Southeastern University, Davie, FL 33328, USA; al2059@mynsu.nova.edu; 6Department of Pathology and Laboratory Medicine, Oregon Health & Science University, Portland, OR 97239, USA; alrohaib@ohsu.edu (A.A.); mohebnas@ohsu.edu (M.M.); 7Biostatistics Shared Resource, Oregon Health & Science University, Portland, OR 97239, USA; bassale@ohsu.edu; 8Knight Cancer Institute—CEDAR, Oregon Health & Science University, Portland, OR 97239, USA; 9Division of Hematology and Medical Oncology, School of Medicine, Oregon Health & Science University, Portland, OR 97239, USA; kardosh@ohsu.edu (A.K.); cheem@ohsu.edu (E.Y.C.); benschk@ohsu.edu (K.G.B.); 10Portland VA Medical Center—Medical Oncology, Portland, OR 97239, USA; 11Dartmouth-Hitchcock Norris Cotton Cancer Center, Department of Radiation Oncology, Lebanon, NH 03756, USA; Charles.R.Thomas.Jr@Dartmouth.edu; 12Division of Surgical Oncology, Oregon Health & Science University, Portland, OR 97239, USA; mayos@ohsu.edu

**Keywords:** neoadjuvant therapy, pancreatic ductal adenocarcinoma, perioperative complications

## Abstract

**Simple Summary:**

The role of neoadjuvant therapy in pancreatic cancer is poorly defined. Our results show improved overall survival in patients who received neoadjuvant therapy, driven by improved chemotherapy delivery, with no apparent increase in early or late perioperative complications.

**Abstract:**

The role of neoadjuvant chemoradiotherapy and/or chemotherapy (neoCHT) in patients with pancreatic ductal adenocarcinoma (PDAC) is poorly defined. We hypothesized that patients who underwent neoadjuvant therapy (NAT) would have improved systemic therapy delivery, as well as comparable perioperative complications, compared to patients undergoing upfront resection. This is an IRB-approved retrospective study of potentially resectable PDAC patients treated within an academic quaternary referral center between 2011 and 2018. Data were abstracted from the electronic medical record using an institutional cancer registry and the National Surgical Quality Improvement Program. Three hundred and fourteen patients were eligible for analysis and eighty-one patients received NAT. The median overall survival (OS) was significantly improved in patients who received NAT (28.6 vs. 20.1 months, *p* = 0.014). Patients receiving neoCHT had an overall increased mean duration of systemic therapy (*p* < 0.001), and the median OS improved with each month of chemotherapy delivered (HR = 0.81 per month CHT, 95% CI (0.76–0.86), *p* < 0.001). NAT was not associated with increases in early severe post-operative complications (*p* = 0.47), late leaks (*p* = 0.23), or 30–90 day readmissions (*p* = 0.084). Our results show improved OS in patients who received NAT, driven largely by improved chemotherapy delivery, without an apparent increase in early or late perioperative complications compared to patients undergoing upfront resection.

## 1. Introduction

Pancreatic cancer is estimated to cause more than 430,000 deaths per year worldwide, ranking as the seventh leading cause of cancer deaths worldwide and third in the United States [1,2,3]. The standard of care for the fraction of patients who present with technically resectable cancer is curative-intent resection with perioperative systemic chemotherapy, with chemoradiation often reserved for patients with a positive surgical margin or lymph node involvement [4]. Optimal sequencing of adjuvant therapies with surgery remains an open question. To date, the most promising outcomes for patients with resectable disease were reported in the PRODIGE 24-ACCORD trial, in which patients who received adjuvant combination chemotherapy with modified FOLFIRINOX demonstrated a median overall survival (OS) of 54.4 months, compared to 35.0 months with gemcitabine alone [5]. This dramatic survival improvement is primarily attributed to the enhanced activity of combination chemotherapy. In practice, a mere 50% of patients receive adjuvant chemotherapy after surgery, due, in part, to the morbidity of pancreatectomy, which can disqualify or delay patients from receiving adjuvant chemotherapy [6,7]. Thus, the benefits of adjuvant combination chemotherapy are not available to approximately half of all patients who undergo upfront surgery.

As has been successfully demonstrated in rectal and esophageal cancer [8,9], a neoadjuvant approach has the advantages of delivering systemic therapy to a greater proportion of patients, selecting for patients with favorable biology, and downstaging to improve R0 resection rates [10,11]. Despite these potential advantages, there is no international consensus on neoadjuvant therapy (NAT) [12]. A single randomized trial, the PREOPANC-1 trial (NTR3709), compared adjuvant to neoadjuvant/perioperative chemotherapy. Although the trial did not meet its primary survival endpoint, it demonstrated improved disease-free survival, locoregional failure-free interval, and reduced pathologic node involvement [13]. Additionally, numerous smaller retrospective studies show a benefit of NAT, specifically improved R0 resection and nodal positivity rates [14,15,16]. Studies consistently show no increase in early perioperative complications following NAT, although no study, to our knowledge, has comprehensively assessed the impact of NAT on late (>30 day) surgical toxicities [17,18,19,20,21,22,23]. In the absence of clear level 1 data, there remain uncertainties regarding the efficacy and toxicity of this approach.

In this study, we utilize a robust, single-institution, highly granular dataset of patients who underwent curative-intent resection for PDAC, to explore the effects of NAT on early and late surgical complications, survival, and the relationship between these outcomes and the delivery of the intended perioperative systemic therapy. We hypothesized that patients who underwent NAT would have improved systemic therapy delivery, and have comparable survival and early and late perioperative complications compared to patients undergoing upfront resection.

## 2. Materials and Methods

This is an IRB-approved retrospective study of 314 consecutive patients with PDAC who underwent curative-intent resection at a National Cancer Institute Comprehensive Cancer Center within an academic quaternary referral center between 2011 and 2018. Data were abstracted from the electronic medical record using an institutional cancer registry and National Surgical Quality Improvement Program (NSQIP) databases, supplemented with chart review.

Neoadjuvant therapy was delivered at the discretion of the multidisciplinary tumor board, who generally recommended neoadjuvant therapy for patients with borderline resectable disease, CA19-9 > 1000 U/mL, regional nodal involvement, or otherwise bulky primary tumors. Borderline resectable was defined using National Comprehensive Cancer Network guidelines [24]. Of note, 57 patients were initially categorized as “resectable” based on tumor board consensus and CT staging, though at the time of surgery they had large vessel encroachment requiring venous and/or vascular resection. Choice of NAT, including neoadjuvant chemotherapy (neoCHT) or chemoradiotherapy (neo-chemoRT), and type of chemotherapy was up to the discretion of the patients’ oncologists. Generally, neo-chemoRT was integrated in patients who had persistent vascular involvement after neoCHT. Patients who had disease progression during NAT that precluded resection were not captured by this data query. Adjuvant chemotherapy or RT was defined as post-operative therapy in the absence of progressive or metastatic disease at the time of treatment onset. All patients were recommended to receive at least 6 months of chemotherapy, neoadjuvant or adjuvant, unless deemed medically unfit. Adjuvant RT was delivered after chemotherapy in non-metastatic patients with positive margins or nodal disease on final pathology. Intraoperative radiation therapy was utilized in select cases when patients were at high risk for positive margin. Patients who received more than four months of neoCHT were excluded from adjuvant CHT analyses, as these patients were less likely to be recommended adjuvant CHT. Chemotherapy delivered after disease recurrence/progression was not considered adjuvant. Post-operative surveillance imaging was obtained per NCCN guidelines, which includes a post-operative baseline study prior to adjuvant CHT (~1–2 months post-operatively), if clinically indicated, and otherwise in 3–6 month intervals.

To score post-operative complications, standard Clavien–Dindo classification was used, and all post-operative complications were strictly defined using NSQIP definitions, including pancreatectomy-targeted variables. Major post-operative complications were defined as Clavien–Dindo grade IIIa or higher [25]. Post-operative pancreatic fistulas (POPF) were only considered if categorized as clinically relevant Bassi grade B or C. Specifically, grade B was defined as drains either left in place > 3 weeks or repositioned through endoscopic or percutaneous procedures. Grade C post-operative pancreatic fistula was scored as those that require reoperation or lead to single or multiple organ failure and/or mortality attributable to the pancreatic fistula. Biochemical POPF (grade A) was not included as per 2016 update of International Study Group (ISGPS) guidelines [26]. Readmissions were only considered if complications were direct or indirect sequela of cancer-directed therapy. Readmissions clearly unrelated, such as trauma, were not coded.

Pathologic grading of response to NAT was performed according to Modified Ryan Scheme for Tumor Regression Score; G0 = no viable cancer cells (complete response), G1 = single cells or rare small groups of cancer cells (near complete response), G2 = residual cancer with evident tumor regression, but more than single cells or rare small groups of cancer cells (partial response), and G3 = extensive residual cancer with no evident tumor regression (poor or no response) [27]. All pathologic records were reviewed by board-certified pathologists or fellows.

Demographics and clinical characteristics were analyzed. Descriptive statistics were presented as frequencies and percentages for categorical variables. Continuous variables were expressed as mean standard deviation. We used chi-square or Fisher’s exact test to compare categorical variables between the treatment groups. The Wilcoxon rank-sum test was used compare median differences in total months of CHT and hospital LOS between patients who received NAT versus no NAT. Overall survival was estimated using the Kaplan–Meier method. Differences in survival distribution between groups were assessed using the log-rank test. The Cox Proportional Hazards model was used for multivariable analysis of OS. Hazard ratios (HR) and their 95% confidence intervals were reported. The criterion for statistical significance was *p* < 0.05. The data were analyzed using SAS (Statistical Analysis System) version 9.4.

## 3. Results

### 3.1. Demographics and Clinicopathologic Characteristics

A total of 314 patients underwent curative-intent resection, 71 (22.6%) of whom were classified as borderline resectable (bR-PDAC), with a median follow-up of 24.1 months. Further, 220 of 314 patients (70.0%) were pT0-2, 94 (30.0%) pT3-4, 78 (24.8%) pN0, 234 (74.5%) pN1-2, 2 (0.01%) pNx, and all were M0. Eighty-one (25.8%) patients received NAT, of whom 71 patients (87.7%) had bR-PDAC and 10 (12.3%) had upfront resectable disease (uR-PDAC). Of the 293 patients who did not receive full-course neoadjuvant CHT, 200 (68.5%) received adjuvant CHT. The overall R0 resection rate was 79.3%, and 9.6% of patients received adjuvant RT. In addition, 37.2% of all the patients received full-course (6+ months) CHT, either neoadjuvant or adjuvant. Additional patient characteristics are referenced in Table 1.

The most common neoCHT were nab-paclitaxel/gemcitabine (39.5%) or FOLFIRINOX (45.7%), and the most common adjuvant CHT was single-agent gemcitabine (49.5%), followed by gemcitabine/capecitabine (19.1%), nab-paclitaxel/gemcitabine (12.3%), and FOLFIRINOX (8.3%). Further, 74.5% of the patients received any CHT, 63.8% received 3+ months CHT, and 37.6% of the patients completed 6+ months CHT (Table 2).

### 3.2. Association of NAT with Oncologic Outcomes and Pathologic Response

The median OS was 22.5 months (Figure 1a), with node-negative (31.1 vs. 20.1 months; *p* = 0.002), margin-negative (28.1 vs. 12.5 months, *p* < 0.001), and tumor grade 1 or 2 disease (26.0 vs. 14.3 months; *p* < 0.001) portending a more favorable prognosis. The median OS was significantly improved in patients who received NAT compared to those who did not (28.6 vs. 20.1 months; Figure 1b *p* = 0.01), with 88% of those receiving NAT having bR-PDAC.

Notably, all bR-PDAC patients received NAT, compared to 4.1% of those with uR-PDAC. Compared to upfront surgery, NAT was not associated with improved R0 resection rates (75.0% with NAT vs. 81.4% without; *p* = 0.23). Patients who underwent NAT required higher rates of vascular resection (48.0% vs. 23.8%; *p* < 0.001), suggesting the presence of more locally advanced disease.

Specifically, neo-chemoRT did appear to have significant oncologic activity on pathologic analysis. The addition of neo-chemoRT to neoCHT was associated with improvements in pathologic response, with 92.9% of the patients having any treatment response compared to 50.0% in the patients treated with neoCHT alone (*p* < 0.001) and a near-complete or complete response (pCR) in 50.0% of those who received neo-chemoRT, compared to 16.7% following neoCHT alone (*p* = 0.004, Figure 2). The median OS in near-complete or complete responders was numerically higher, but not statistically significant (60.6 (near complete or complete response) vs. 21.8 months (partial or no response); *p* = 0.07). Patients for whom pathologic response data were unavailable were excluded from the analysis.

### 3.3. Association of Neoadjuvant Therapy with Early and Late Surgical Complications

NAT was not associated with an increase in severe early post-operative complications (25.0% with NAT vs. 19.9% without; *p* = 0.34), length of stay (LOS) (median 10d vs. 10d; *p* = 0.87), 30-day readmission rates (28.2% with NAT vs. 18.2% without; *p* = 0.06), late leak rates (7.7% with NAT vs. 4.1% without; *p* = 0.23), or 30–90 day readmission rates (20.5% with NAT vs. 12.3% without; *p* = 0.08), as shown in Table 3. Of note, the rates of vascular reconstruction were significantly higher in patients who underwent NAT (25% (no NAT) versus 47.4% (NAT); *p* < 0.001).

As compared to the patients who received neoCHT alone (*n* = 48), the patients who received neo-chemoRT (*n* = 33) were not more likely to experience a major early complication (28.1% with neo-chemoRT vs. 20.8% without; *p* = 0.29), including if they underwent vascular resection (44.4% with neo-chemoRT versus 31.7% without; *p* = 0.30 (Appendix A). Neo-chemoRT was associated with significantly decreased rates of POPF (0% with neo-chemoRT vs. 14% without; *p* = 0.012) and post-operative sepsis (3% with neo-chemoRT vs. 19% without; *p* = 0.02). Both of these effects appear to be specific to patients who received neo-chemoRT, as neoCHT alone was not associated with decreased POPF (14% vs. 15%; *p* = 0.82) or post-operative sepsis (20% vs. 15%; *p* = 0.44). Late 30–90 day readmissions (25.8% with neo-chemoRT vs. 17.0% without; *p* = 0.35) or late leaks (3.2% with neo-chemoRT vs. 10.6% without; *p* = 0.23) were not different between those who received neo-chemoRT versus neoCHT alone.

A persistent concern amongst oncologists is the risk of late post-operative complications and subsequent readmissions related to chemoradiation; the specific etiologies for late readmissions in patients who received NAT (*N* = 16 of 81 patients, 19.8%) were explored. Three patients were found to have severe wound complications, and only one patient suffered a directly attributable complication of neo-chemoRT—late radiation enteritis complicated by recurrent gastrointestinal bleeding.

### 3.4. Association between Systemic Therapy and Oncologic Outcomes

The patients who received CHT exhibited improved OS compared to those who did not (28.1 months with CHT vs. 8.3 months with no CHT, *p* < 0.001; Figure 3a). The median OS improved with each month of chemotherapy delivered (HR = 0.81 per month CHT, 95% CI 0.76–0.86, *p* < 0.001). The patients who received at least 3 months of CHT had significantly improved median OS (28.6 months vs. 10.1 months, *p* < 0.001; Figure 3b), and those who received 6+ months of CHT had improved median OS (36.3 months (6+ months CHT) vs. 13.5 months (<6 months CHT), *p* < 0.001; Figure 3c). On average, patients treated with neoCHT received a longer duration of systemic therapy than those who had upfront surgery (5.0 ± 2.3 months vs. 3.3 ± 2.8 months, *p* < 0.001; Figure 3d).

The OS was compared between the NAT and no NAT groups in patients who received 3+ or 6+ months of chemotherapy. In those who received 3+ months of chemotherapy, there were no significant differences in survival between the NAT and no NAT groups (29.6 months (no NAT) vs. 31.1 months (NAT), *p* = 0.47). Similarly, in those who received 6+ months of chemotherapy, there were no significant differences in OS (36.9 months (no NAT) vs. 37.7 months (NAT), *p* = 0.78).

Patients suffering a major surgical complication were significantly less likely to receive any adjuvant CHT (41.0% with a major complication vs. 81.1% without; *p* < 0.0001), and had poorer median OS (12.2 months with a major complication vs. 24.0 months without, *p* = 0.002). On average, patients without severe complications received adjuvant chemotherapy for a longer duration (3.6 ± 2.7 months vs. 1.2 ± 2.1 months, *p* < 0.001). Accordingly, major complications significantly reduced the proportion of patients receiving ≥3 months of adjuvant CHT (20.0% vs. 66.7%, *p* < 0.001), and only 10.0% of patients suffering a major complication completed a full 6-month course of perioperative CHT compared to 39.3% who did not (*p* < 0.001).

## 4. Discussion

The potential benefits of NAT in PDAC include better patient selection by disease biology, improved systemic therapy compliance, disease downstaging, and improved R0 resection rates. Our results show comparable outcomes, surgical complications, and improved CHT delivery in those who underwent NAT. We observed that patients who received NAT, despite having more locally advanced disease, had superior survival rates compared to those who did not. In contrast to the PREOPANC-1 trail, the R0 resection rates were not improved with NAT in this cohort; however, the increased vascular reconstruction rate and more locally advanced nature of the disease among patients who received NAT at this institution confound interpretation of this relationship.

An important question addressed and expanded upon by the present study is the perioperative safety of NAT up to 90 days postoperatively, as NSQIP databases routinely only track patients over the first 30 days following surgery. The NAT cohort was selected for more locally advanced disease, as evidenced by the higher rates of vascular reconstruction and bR-PDAC; however, despite this difference, patients in the NAT group suffered equivalent rates of major early and late perioperative complications. Radiation, specifically, is known to cause irreversible DNA damage in non-target tissues, leading to fibrosis and wound healing complications, though the clinical implications of these late toxicities in patients with PDAC are unclear [28]. A 2014 ACS-NSQIP analysis of 4416 patients undergoing PDAC resection, 200 of whom underwent neoRT, showed equivalent 30-day perioperative complication rates between those who underwent neoRT and those who did not; however, it was suggested that neo-chemoRT may increase operative complexity and complications [29]. Our data, in contrast, showed a significant decrease in early POPF and sepsis, with no significant difference in both early and late complications and hospital readmission rates. Upon detailed exploration, only one patient who received neo-chemoRT suffered late radiation-induced enteritis, and those who received neoCHT alone did not experience increased complication rates.

We suspect there was improved overall chemotherapy delivery in patients who underwent NAT because the post-operative complications in patients with uR-PDAC impeded planned adjuvant CHT. Seventy-three patients who did not receive any NAT ultimately did not receive adjuvant CHT, due to rapid post-operative disease progression, surgical complications, or a decline in functional status. These are the key patients that we feel would have benefited from early integration of systemic chemotherapy or chemoradiotherapy. Though the perioperative complications were comparable between the NAT and no NAT groups, the treatment course of those who received full-course neoadjuvant chemotherapy was not impacted negatively. Our work supports previous studies, which show that a longer duration of CHT is strongly associated with improved survival [30,31]. We observed that major surgical complications were associated with both markedly decreased CHT delivery and poorer OS. The patients who suffered a CD G3+ complication, 21% of the patients, were half as likely to receive adjuvant chemotherapy, and exhibited an 11.8 month decrease in median OS. Furthermore, these patients were roughly six-fold less likely to complete their recommended course of 6 months of systemic therapy. The patients who received neoCHT received an average of an additional 2 months of chemotherapy. Given that we did not observe increased complication rates in the NAT cohort with increased overall CHT delivery, the authors favor NAT to be the major driving factor of improved survival outcomes.

Improved survival in patients who underwent NAT is also due, in part, to the negative selection of patients with a decline in performance status or progressive disease during NAT. These patients did not go to surgery and, unfortunately, could not be captured within this data query. As shown in the PREOPANC-1 trial, nearly 40% of the patients randomized to the NAT arm did not undergo curative-intent resection, due to metastatic disease identified at staging laparoscopy (10.9%), progression prior to NAT (3.3%), or disease progression after starting NAT (10.9%), while major NAT complications precluding surgery occurred in only one patient (0.8%). Further, in an effort to elucidate the unique therapeutic benefits of NAT, we compared OS in NAT versus no NAT groups in those who received 3+ or 6+ months of chemotherapy. By limiting the analysis to those groups, patients were positively selected in both the NAT and no NAT groups to have improved OS. There were no OS differences observed between NAT and no NAT in either case, which suggests that the therapeutic benefit is similar in the neoadjuvant and adjuvant setting. However, NAT has the aforementioned benefits of improved chemotherapy delivery, and, thus, showed improved OS in the entire cohort. In the absence of improved prognostic tools, the “test of time” that NAT provides is also currently the best method for identifying patients unlikely to benefit from curative-intent resection.

Although this small, selected cohort lacks power to address whether neo-chemoRT improves clinical outcomes, neo-chemoRT did exhibit significant anti-tumoral activity, with markedly higher pathologic response rates, compared to those who underwent neoCHT alone. Consistent with other published literature, there was a signal that patients with pCR or near-complete responses have a survival advantage compared to those with partial or poor responses [32]. The role for neo-chemoRT in potentially resectable disease remains controversial; however, these data add to the growing literature suggesting that RT is locally efficacious against PDAC [33,34,35].

This study is limited by its retrospective, single-institution nature. Importantly, patients progressing or with complications during neoadjuvant therapy, precluding curative-intent surgery, were unfortunately not able to be captured through data queries. Therefore, it is possible that the included patients who received NAT may have had more favorable biology, despite having more locally advanced disease at diagnosis. Further, this study does not have the power to make a meaningful comparison between patients with uR-PDAC who received NAT and those who did not.

## 5. Conclusions

In summary, the patients with PDAC who received NAT had improved overall survival compared to those who did not, which appeared to be driven by patient selection and improved duration of chemotherapy delivery, with no apparent increase in early or late severe complications or readmissions. Though the clinical outcomes could not be compared, neo-chemoRT markedly improved pathologic response rates, suggesting significant anti-tumoral activity. Based on this study and mounting evidence supporting an NAT approach, phase III trials are required to further explore the utility of NAT in all patients with both borderline and resectable PDAC.

## Figures and Tables

**Figure 1 cancers-14-00609-f001:**
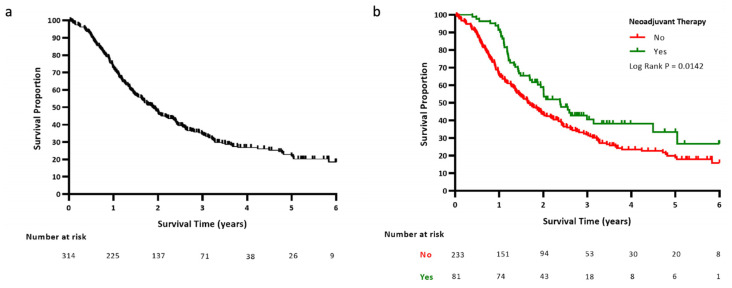
Kaplan–Meier OS curves for all patients combined (**a**) and for those who received NAT versus no NAT (**b**).

**Figure 2 cancers-14-00609-f002:**
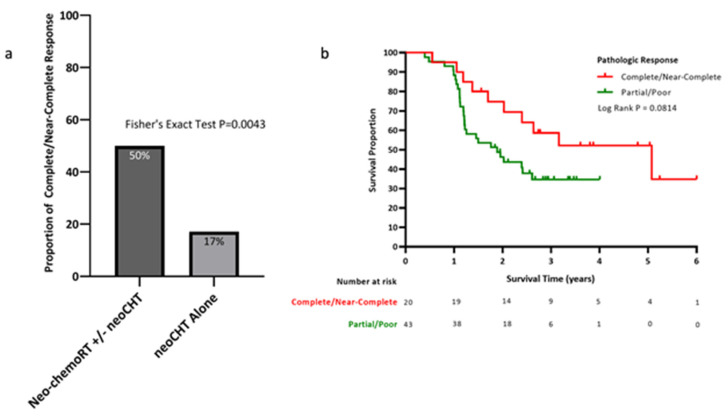
Neo-chemoRT is associated with increased pathologic response rates compared to neoCHT alone. Figure demonstrates (**a**) increased rates of “complete or near-complete response” in those who received neo-chemoRT +/− neoCHT versus neoCHT alone and (**b**) improved median OS in complete/near-complete responders.

**Figure 3 cancers-14-00609-f003:**
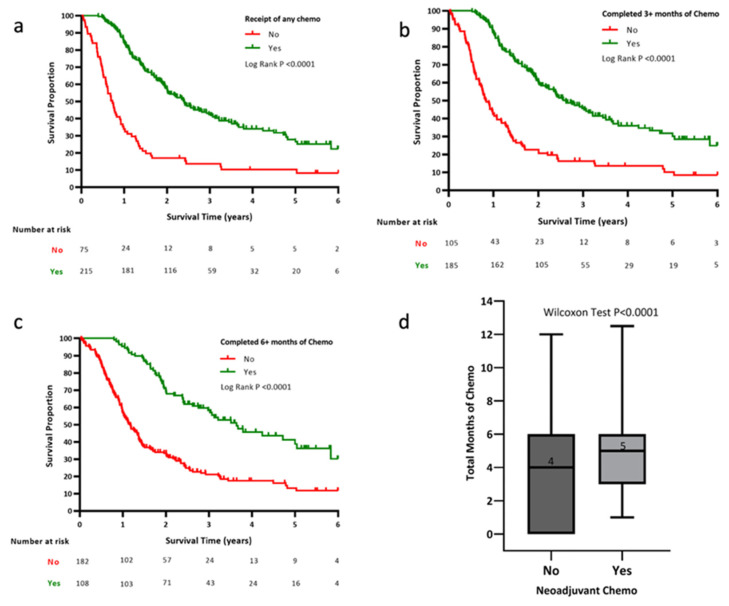
Receipt of systemic therapy is strongly associated with improved OS. Kaplan–Meier OS for patients who received any CHT (**a**), 3+ total months of CHT (**b**), and 6+ total months of CHT (**c**). Log-rank *p*-values are displayed. (**d**) A box plot comparison of the total months of CHT in those who received neoCHT versus those who did not.

**Table 1 cancers-14-00609-t001:** Comparison of patient characteristics between those who received NAT and those who did not.

Variable	Overall (N = 314)	No NAT (N = 233)	NAT (N = 81)	*p*
**Age**				
<65 year	134 (42.7)	93 (39.9)	41 (50.6)	0.09
≥65 year	180 (57.3)	140 (60.1)	40 (49.4)	
**Biological sex**				
Female	140 (44.6)	105 (45.1)	35 (43.2)	0.77
Male	174 (55.4)	128 (54.9)	46 (56.8)	
**Body Mass Index (BMI)**				
BMI 18–24.9	112 (35.8)	83 (35.8)	29 (35.8)	0.49
BMI 25–29.9	107 (34.2)	83 (35.8)	24 (29.6)	
BMI ≥ 30	94 (30)	66 (28.5)	28 (34.6)	
**Stage on presentation**				
uR-PDAC	243 (77.4)	233 (100)	10 (12.4)	<0.001 *
bR-PDAC	71 (22.6)	0 (0)	71 (87.7)	
**Diabetes at diagnosis**				
Yes	99 (31.5)	74 (31.8)	25 (30.9)	0.88
No	215 (68.5)	159 (68.2)	56 (69.1)	
**Neoadjuvant Treatment**				
Chemotherapy alone	48 (15.3)	0 (0)	47 (59.3)	NA
Chemotherapy + chemoradiotherapy	31 (9.9)	0 (0)	31 (38.3)	
Chemoradiotherapy alone	2 (0.6)	0 (0)	2 (2.5)	
None	233 (74.2)	233 (100)	0 (0)	
**Operation Performed**				
Pylorus-preserving Whipple	124 (39.5%)	98 (42.0%)	26 (32.1%)	0.45
Standard Whipple	125 (39.8%)	86 (36.9%)	39 (38.1%)	
Radical anterograde modular pancreatosplenectomy (RAMPS) +/− Appleby	45 (14.3%)	35 (15.0%)	10 (12.3%)	
Distal pancreatectomy +/− splenectomy	17 (5.4%)	12 (5.2%)	5 (6.2%)	
Operative report unavailable	3 (1.0%)	2 (1.0%)	1 (1.2%)	
**Resection margin**				
R0	249 (79.3)	189 (81.1)	60 (74.1)	0.18
R1	65 (20.7)	44 (18.9)	21 (25.9)	
**Grade Differentiation**				
G1	10 (3.6)	7 (3.1)	3 (5.5)	0.40 ^(a)^
G2	166 (59.5)	137 (61.2)	29 (52.7)	
G3	101 (36.2)	80 (35.7)	21 (38.2)	
G4	2 (0.7)	0 (0)	2 (3.6)	
**Pancreatectomy vascular resection**				
Vein	77 (24.8)	50 (21.6)	27 (34.6)	<0.001 *
Artery	9 (2.9)	7 (3)	2 (2.6)	
Both	10 (3.2)	2 (0.9)	8 (10.3)	
None	214 (69)	173 (74.6)	41 (52.6)	
**Type of first recurrence**				
Local	55 (17.6)	40 (17.2)	15 (18.8)	0.30
Distant	140 (44.7)	103 (44.2)	37 (46.3)	
Regional	12 (3.8)	10 (4.3)	2 (2.5)	
No Recurrence	125 (39.9)	97 (41.6)	28 (35)	

^(a)^ G1 & G2 and G3 & G4 were combined to estimate this *p* value. * *p*-value < 0.05.

**Table 2 cancers-14-00609-t002:** Comparison of patterns of care between those who received NAT and those who did not.

Variable	Overall (N = 314)	No NAT (N = 233)	NAT (N = 81)	*p*
**Type of Neoadjuvant Chemotherapy**				
Nab-paclitaxel/gemcitabine			32 (39.5)	NA
Modified oxaliplatin, leucovorin, irinotecan and fluorouracil (mFOLFIRINOX)			37 (45.7)	
Other			12 (14.8)	
**Receipt of Adjuvant Chemotherapy**				
Yes	204 (65.2)	160 (68.7)	44 (55)	0.03 *
No	109 (34.8)	73 (31.3)	36 (45)	
**Type of Adjuvant Chemotherapy**				
Gemcitabine	101 (49.5)	90 (56.3)	11 (25)	<0.001 *
Gemcitabine/Capecitabine	39 (19.1)	32 (20)	7 (15.9)	
Gemcitabine/Nab-paclitaxel	25 (12.3)	14 (8.8)	11 (25)	
mFOLFIRINOX	17 (8.3)	13 (8.1)	4 (9.1)	
Other	22 (10.8)	11 (6.9)	11 (25)	
**Received any Chemotherapy**				
Yes	216	143 (66.2)	73 (98.6)	<0.001 *
No	74	73 (33.8)	1 (1.4)	
**Completed 3+ months of Chemotherapy**				
Yes	185 (63.8)	123 (56.9)	62 (83.8)	<0.001 *
No	105 (36.2)	93 (43.1)	12 (16.2)	
**Completed 6 months of Chemotherapy**				
Yes	109 (37.6)	75 (34.7)	34 (45.9)	0.13
No	181 (62.4)	141 (65.3)	40 (54.1)	
**Adjuvant Radiotherapy**				
Yes	31 (9.9)	22 (9.4)	9 (11.1)	0.66
No	283 (90.1)	211 (90.6)	72 (88.9)	

* *p*-value < 0.05.

**Table 3 cancers-14-00609-t003:** Comparison of hospital LOS, early and late complications, and early and late readmission rates between patients who underwent NAT versus no NAT.

Variable	Overall (N = 314)	No NAT (N = 233)	NAT (N = 81)	*p*
**Hospital Length of Stay**				
Mean (SD)	15.1 (17.2)	14.8 (16.4)	18 (15.9)	0.87
Range	2–200	3–200	2–138	
**C-D Grade 3+ Complications**				
Yes	66 (21.2)	46 (19.9)	20 (25)	0.34
No	245 (78.8)	185 (80.1)	60 (75)	
**30-day Bile Leaks**				
Yes	22 (7.2)	18 (8)	4 (5.1)	0.39
No	283 (92.8)	208 (92)	75 (94.9)	
**Non-pancreatic Leak**				
Yes	37 (11.9)	32 (13.9)	5 (6.3)	0.07
No	273 (88.1)	198 (86.1)	75 (93.8)	
**Post-operative wound disruption**				
Yes	60 (19.2)	42 (18.1)	18 (22.5)	0.39
No	252 (80.8)	190 (81.9)	62 (77.5)	
**Post-operative Sepsis**				
Yes	55 (17.6)	47 (20.3)	8 (10)	0.04 *
No	257 (82.4)	185 (79.7)	72 (90)	
**Post-operative Pneumonia**				
Yes	18 (5.8)	12 (5.2)	6 (7.5)	0.42
No	294 (94.2)	220 (94.8)	74 (92.5)	
**Late (>30 day) leak**				
Yes	15 (5)	9 (4.1)	6 (7.7)	0.23
No	284 (95)	212 (95.9)	72 (92.3)	
**30 day mortality**				
Yes	11 (3.5)	9 (3.9)	2 (2.5)	0.74
No	299 (96.5)	221 (96.1)	78 (97.5)	
**30 day readmission rate**				
Yes	62 (20.8)	40 (18.2)	22 (28.2)	0.06
No	236 (79.2)	180 (81.8)	56 (71.8)	
**30–90 day readmission rate**				
Yes	43 (14.4)	27 (12.3)	16 (20.5)	0.08
No	255 (85.6)	193 (87.7)	62 (79.5)	
**Rate of Vascular Reconstruction**				
Yes	95 (30.7)	58 (25)	37 (47.4)	<0.001 *
No	215 (69.4)	174 (75)	41 (52.6)	

* *p*-value < 0.05.

## Data Availability

The data presented in this study are available on request from the corresponding author. The data are not publicly available due to ongoing studies and for patient privacy.

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
