# Peer review of "Neoadjuvant Therapy Is Associated with Improved Chemotherapy Delivery and Overall Survival Compared to Upfront Resection in Pancreatic Cancer without Increasing Perioperative Complications"

_cancers, 2022, doi:10.3390/cancers14030609_

Round 1

Reviewer 1 Report

This retrospective study was aimed to investigate the effect and influence of neoadjuvant chemotherapy in patients with pancreatic cancer, especially on systemic chemotherapy delivery and postoperative complications.

1, In 233 upfront resectable patients of No-NAT group, 50 patients underwent portal vein resection and 7 patients underwent combined arterial resection as shown in Table 1. What were the causes to require combined these vascular resections in 57 patients in the group of No-NAT, despite of up-front  resectable pancreatic cancers at the diagnosis? Authors should clarify the reasons for requiring combined vascular resections in these patients.

2, There were 73 patients who did not receive adjuvant chemotherapy in No-NAT group. Why did they receive adjuvant chemotherapy after surgical resection? Its reasons for these 73 patients should be clarified and mentioned clearly despite no obvious difference in the occurrence and its severity of postoperative complications between NAT-and non-NAT groups.

3, This study strongly addressed that neoadjuvant therapy improved systemic delivery of chemotherapy more than non-NAT group, and showed that more than 3 months chemotherapy delivery and more than 6 months chemotherapy delivery could significantly improved overall survival as shown in Fig 3. If comparison between overall survival NAT and non-NAT groups in more than 3 months and more than 6 months of chemotherapy group are shown, its result would reveal the implication of neoadjuvant chemotherapy more clearly in this study. Authors should the results of its analysis.

4, There were 31 patients who underwent chemoradiotherapy in the NAT regimen, and 47 patients who underwent chemotherapy alone in the NAT regimen. What was the difference of chemotherapy alone and chemoradiotherapy of NAT regimen in effects on postoperative complications and also overall survival?  These results might be more clearly understood to know the effect of neoadjuvant therapy.

Author Response

Thank you for your thoughtful review of our manuscript. We highly value frank, honest feedback as we wish to improve the quality of the paper. We will attempt to address your concerns point by point, and make appropriate changes in the manuscript with line number references in parentheses. Please see the attached file. 

Reviewer 2 Report

The authors retrospectively investigated a clinical role of neoadjuvant therapy (NAT) in 314 patients with pancreatic ductal adenocarcinoma (PDAC) who underwent curative-intent pancreatectomy from 2011 to 2018. Subsequently, implementation of NAT was associated with the improved survival without an apparent increase in peri-operative complications.

Although this manuscript is well written, there is little new information.

Major comments.

  1. On investigating clinical effect of NAT, intention-to-treat analysis should be done in NAT group and upfront surgery group. Otherwise selection bias would affect the survival outcome.
  2. NAT was intensively selected to the patients with highly staged PDAC in this study. In spite of this difference, NAT was associated with improved survival. NAT might exclude surgically unfit patients. The authors should expose total population of NAT and no NAT groups including non-resected patients.
  3. NAT can usually induce high proportion of negative lymph node metastasis and margin-negative resection resulting in low proportion of local recurrence especially in patients who underwent radiation therapy. Please explain higher proportion of local recurrence in NAT group (Table 1).
  4. “upfront resectable (ur)” can be easily misrecognized as unresectable in general. Please change to “R”.
  5. Please spell out abbreviation used in Tables.
  6. In Table 3, 81 of hospital stay in NAT group may be 18?
  7. There was a significant difference in surgical reconstruction between two groups in Table 3. What does it mean? Please explain it in the Results and state the reason of this difference in the discussion.

I appreciate for giving me an opportunity to review this article.

Author Response

Thank you for your very thoughtful and detailed review of our manuscript. We appreciate the opportunity to improve our work. We understand that a key limitation of the study is that the NAT cohort is selected to include patients who's cancer may behave biologically different than uR-PDAC patients. Though a paired-matched analysis including patients who started NAT, though did not make it to surgery would be ideal, it would not be possible with our current database query capabilities. We have made our best attempt at being transparent about this limitation and interpreting the results in this context, largely focusing on the safety of NAT, including late complications, as well as improved chemotherapy delivery. We hope that you will consider our manuscript further based on the changes made. We greatly appreciate your time and consideration. 

Please see the attached point by point response with referenced line numbers for the revised manuscript. 

Reviewer 3 Report

I have read with interest the manuscript by Dr Grossberg and colleagues that compare overall survival and therapy responses in PDAC patients who underwent neoadjuvant therapy (NAT) compared to patients undergoing upfront surgical resection. They report an enhanced overall survival in patients who received neoadjuvant therapy (underlying cause being efficient delivery of chemotherapy) without an associated increase in perioperative complications. The study is well balanced and patient demographics have been discussed in detail. The use of statistical analyses is appropriate. Overall, the findings are clinically relevant and will add valuable information regarding survival benefit in PDAC patients.

Author Response

Thank you for your thoughtful review of our manuscript. We are glad that you felt the study design and execution was strong, and the results/conclusions were well-written. We hope you enjoy the final iteration as well. 

Reviewer 4 Report

Your study described a promised therapy in the foggy topic of pancreatic adenocarcinoma. 

The work was well written and all the subparts were subsequent. 

I have no questions 

Author Response

Thank you for your thoughtful review of our manuscript that we have worked hard to bring to publication. We hope you will enjoy the final iteration after making modifications based on others' feedback. 

Round 2

Reviewer 1 Report

Nothing to be further requested